# Scaling Controllable Modeling Via Self-Evolving Feature Engineering

## Abstract

Machine learning faces a fundamental dilemma: models that achieve high predictive performance are typically opaque, while models that provide control, the ability to understand and guide their outcomes, often sacrifice accuracy for transparency. This performance-control trade-off constrains ML adoption in critical domains where both capabilities are essential. Practitioners have historically addressed this challenge through manual feature engineering, embedding domain expertise into models to achieve reasonable accuracy while retaining some degree of control. However, this process is costly, time-consuming and limited by human expertise, restricting scalability. We present FEST (Feature Engineering with Self-evolving Trees), a framework for automated controllable modeling that replaces manual feature design with an iterative, self-evolving process. FEST leverages large language models (LLMs) as feature discovery engines to generate plausible features from observational data by analyzing contrasting samples. Next, these features are semantically clustered, deduplicated and validated for predictive performance using interpretable decision trees. The evolving trees refine feature sets over iterations, producing human-readable decision rules that practitioners can inspect, modify and intervene upon, thus providing both accuracy and control. To demonstrate FEST's effectiveness in bridging the performance-control gap, we evaluate it against traditional interpretable models, neural networks, and LLM classifiers across diverse real-world tasks in social science, NLP, and marketing domains. We also introduce GLoRE, a controlled synthetic benchmark, designed to test to test a model's ability to deduce outcomes from complex logical rule relationships embedded in natural language, with true features and their relationships unknown to LLMs. FEST recovers all of the target features. These results show that automated, self-evolving feature engineering can make controllable modeling practical at scale, reducing reliance on costly manual design while narrowing the long-standing divide between performance and control in machine learning.

## 1 Introduction

Machine learning faces a fundamental dilemma that has persisted since its inception: models that achieve high predictive performance are often opaque, while models that provide control, the ability to understand and guide their outcomes, tend to underperform. Here, performance means a model's ability to make accurate, generalisable predictions, whereas control refers to the ability to understand and guide a model's outcomes. For instance, consider rocket trajectory modeling in aerospace engineering: scientists achieve both precise predictions (performance) and complete understanding of how thrust, launch angle, and atmospheric drag affect outcomes (control). In contrast, machine learning practitioners must choose between high-performing but opaque neural networks and interpretable but underperforming traditional models, creating a persistent challenge that constrains ML adoption in critical applications, such as healthcare diagnostics, financial lending, and policy decisions. This tension is particularly pronounced in domains where both performance and control are essential, such as healthcare diagnostics, financial lending, and policy decisions.

Why does control matter so fundamentally in machine learning systems? Beyond regulatory requirements and ethical considerations, control enables practitioners to understand which features drive predictions, modify model behavior predictably and ensure reliable operation in new scenarios Lipton (2018); Rudin (2019). Real-world deployments demonstrate this critical need: credit risk scorecards remain the industry standard at major banks precisely because regulators require transparent decision processes that can be audited and explained Baesens et al. (2003). Similarly, the Framingham Risk Score continues as the global standard for cardiovascular risk assessment because clinicians can

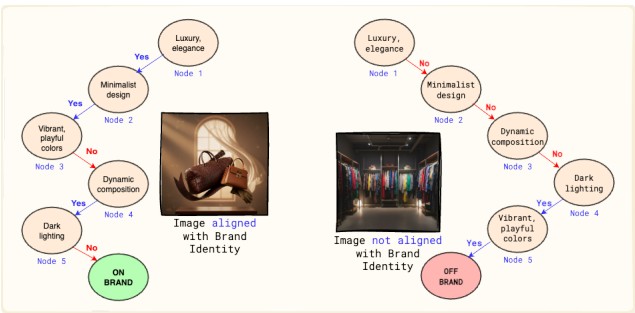

Figure 1: Examples of features and decision rules discovered by FEST for checking alignment of images to brand's visual identity. This produces human-readable decision rules that practitioners can inspect and modify.

understand and trust its additive structure Lloyd-Jones et al. (2004). These systems deliberately sacrifice some predictive power to maintain the control that practitioners require for high-stakes decisions.

Historically, practitioners achieved this essential control through manual feature engineering, the painstaking process of designing input variables that embed domain expertise directly into models. Early systems from the late 1950s onward illustrated this approach: Samuel's checkers program Samuel (1959) used carefully crafted board evaluation features like piece advantage and mobility that human experts could understand and validate, while the MYCIN medical diagnosis system Shortliffe (1976) relied on interpretable clinical features that physicians could trace through logical decision rules. This philosophy extended to practical applications like spam email classification, where models used transparent features such as suspicious word frequencies ("free", "urgent", "click here") and sender metadata (domain reputation, header inconsistencies) to achieve both explainable decisions and reasonable accuracy Sahami et al. (1998). These features served as a bridge between raw signals and human understanding, allowing model decisions to be inspected. This approach provided the control that practitioners needed while delivering acceptable performance.

As machine learning matured, feature engineering remained central. Support Vector Machines Cortes & Vapnik (1995) mapped features into high-dimensional spaces to improve separability through sophisticated kernel methods, yet their performance remained critically dependent on appropriate feature selection and kernel choice, requiring deep domain expertise to achieve optimal results. Decision trees Breiman et al. (1984) offered transparent hierarchical decision rules that could be directly interpreted by domain experts, with each internal node representing a simple threshold test on a single feature, making the entire decision process traceable and modifiable. This property made them particularly valuable in domains requiring regulatory compliance such as medical diagnosis and credit scoring Quinlan (1986). Ensemble methods like Random Forests Breiman (2001) and XGBoost Chen & Guestrin (2016) improved robustness while preserving interpretability through feature importance measures. However, all these algorithmic advances shared a fundamental limitation: their success remained contingent on the quality of manually engineered features.

However, manual feature engineering faces inherent limitations that increasingly constrain performance as machine learning tackles more complex problems. Domain experts are bounded by cognitive limitations, finite knowledge, cultural context and economic constraints Kahneman (2011), creating bottlenecks that become more severe as data complexity increases. As datasets grew from hundreds of features in early applications to thousands in text classification and millions of pixels in computer vision, the cost of hiring domain experts for each new task became economically prohibitive. Even when experts were available, their handcrafted features frequently proved insufficient for competitive performance on complex real-world problems, creating a fundamental tension between the control offered by manual feature engineering and the performance demanded by practical applications.

The deep learning revolution addressed these performance limitations by fundamentally abandoning manual feature engineering in favor of automatic representation learning from raw data. Human expertise was redirected from feature design to architectural innovation. Convolutional neural networks LeCun et al. (2002) learned hierarchical feature detectors directly from pixels, eliminating the need for hand-crafted visual features, while transformer architectures Vaswani et al. (2017) automatically extracted semantic representations from text without requiring linguistic expertise. This architectural revolution achieved dramatic performance gains across vision, language and audio domains by trading control for performance. Thus, practitioners could no longer directly influence

which features models emphasized, understand the reasoning behind predictions, or predict how models would behave in new situations.

Recognizing the critical loss of control in deep learning systems, post-hoc explainability methods were developed to explain model behavior after training. Techniques such as LIME Ribeiro et al. (2016), SHAP Lundberg & Lee (2017), and gradient-based attribution methods Selvaraju et al. (2017) emerged to attribute importance to input features or concepts. However, these approaches face limitations that prevent them from truly restoring control: they can produce misleading results when models rely on spurious correlations Adebayo et al. (2018), lack ground-truth references for verification, and most critically, provide explanation without genuine control. Practitioners can obtain plausible explanations of model behavior without gaining the ability to predict or systematically modify how models will respond to new inputs or scenarios. As Rudin (2019) argues, such explanations can be misleading and should not be relied upon in high-stakes settings. This gap between explanation and control means that explainability research, while valuable for model auditing and retrospective analysis, has not resolved the core challenge of building simultaneously powerful and controllable machine learning systems.

Similarly, reasoning techniques in Large Language Models, while appearing to offer a bridge between performance and explainability, suffer from analogous limitations. Chain-of-Thought (CoT) prompting Wei et al. (2022) and related techniques demonstrate impressive performance improvements on reasoning benchmarks while appearing transparent through step-by-step explanations. However, systematic investigation reveals fundamental faithfulness problems: CoT can generate different reasoning paths for identical inputs across multiple runs Wang et al. (2022), and explanations frequently diverge from the true computations driving predictions, instead rationalizing predetermined answers through seemingly logical but unfaithful post-hoc narratives Barez et al. (2025). Thus, while LLM reasoning appears explainable, it does not provide verifiable control.

We argue that LLMs are more promising in a different role: not as black-box predictors or unreliable explainers, but as *feature discovery engines*. LLMs possess capabilities that could address the fundamental limitations of manual feature engineering. Unlike human experts who are constrained by domain knowledge, cognitive limitations, and economic costs, LLMs are trained on vast amounts of knowledge across diverse domains and demonstrate remarkable pattern recognition abilities that span multiple tasks and domains within a single system Brown et al. (2020). This suggests a path toward scalable automated feature discovery that could overcome the expertise bottlenecks and economic limitations that have constrained manual approaches.

To this end, we present FEST (Feature Engineering with Self-evolving Trees), a novel framework for automated and scalable controllable modeling. FEST couples the pattern recognition capabilities of LLMs with the control of traditional tree-based models through an iterative generate-and-filter process. Given observational data with varying outcomes, FEST prompts LLMs to generate plausible features that distinguish contrasting samples, deduplicates and clusters similar features using semantic embeddings, then trains decision trees to evaluate feature relevance. Trees evolve iteratively, selecting predictive features while discarding weak ones, producing models whose decision logic is expressed as explicit, human-readable rules. This process continues until convergence, building a refined bank of validated features that enable both high performance and control. By combining LLM-based feature discovery with interpretable model validation, FEST reduces reliance on costly manual engineering while narrowing the performance gap between interpretable and black-box models. Our key contributions are summarised below:

- **Scalable feature engineering**: We introduce FEST, a framework that automates and scales feature engineering via a generate–deduplicate–validate cycle, coupling LLM-based feature discovery with self-evolving decision trees.

- **Verifiable control**: We demonstrate that FEST enables *controllable modeling*: Since FEST leverages interpretable decision tree models, predictions can be traced through a series of simple feature-based decisions, providing reproducible and controllable explanations.

- **Maintained performance**: We provide extensive empirical evaluation. FEST achieves competitive performance across real-world tasks across across social science, NLP, marketing domains, and outperforms baselines on majority of the tasks.

- **Synthetic Benchmark**: We synthesise **GLoRE**, a controlled dataset to test model's ability to deduce outcomes from complex logical rule relationships embedded in natural language. We show that FEST recovers the true features in this setting, validating its capability for not just feature discovery but also relationships between features.

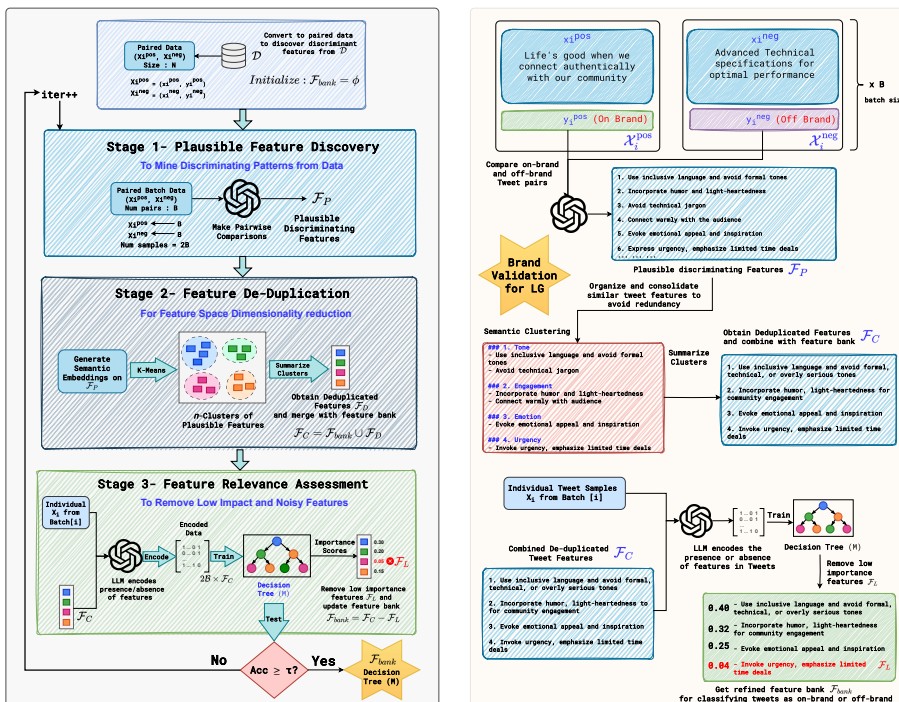

Figure 2: FEST: Feature Engineering with Self-evolving Trees

## 2 METHODOLOGY

FEST operates through an iterative process that extracts interpretable decision features in natural language from raw observational data. The algorithm processes data in expanding batches, continuously refining a feature bank through three core stages: plausible feature discovery, deduplication, and feature relevance assessment, each of which we discuss in detail below. The entire end-to-end pipeline is illustrated in Figure 2. The detailed pseudocode is provided in Algorithm 1.

### 2.1 PAIRWISE COMPARISON FRAMEWORK

**Motivation – From Absolute Assessment to Relative Discrimination**: The central challenge in automated feature engineering is discovering characteristics that distinguish between different types of instances in a dataset. Traditional approaches analyze individual samples in isolation, but this absolute assessment suffers from two fundamental limitations: (1) inability to distinguish between universally present attributes and discriminative features, and (2) dependence on subjective absolute thresholds.

We adopt a relative discrimination approach through pairwise comparisons for two key reasons. First, this approach naturally filters out common attributes and focuses on discriminative features. Consider news headline analysis: absolute assessment might identify "contains numbers" as a relevant feature without considering that numbers appear almost equally in both successful and unsuccessful headlines. Pairwise comparison focuses discovery on features that actually differentiate between contrasting samples, for example, identifying that successful headlines use specific question formats while unsuccessful ones rely on generic statements. Second, Psychology research demonstrates that humans make more consistent judgments when comparing alternatives rather than providing absolute ratings Thurstone (1927). Inspired from this, we hypothesise that pairwise comparisons align better with LLM capabilities for feature discovery. We formally design the pairwise comparison framework as follows:

- Consider a dataset $D = \{(x_i, y_i)\}_{i=1}^N$ where $x_i$ represents the input sample (e.g., text, structured data) and $y_i$ represents the associated outcome measure. The outcome $y_i$ can take different forms depending on the task – (1) Regression: Continuous values (e.g., engagement scores, click rates) (2) Classification: Binary labels (e.g., genuine vs. fake reviews)

- We transform this dataset into pairwise comparisons by constructing comparison pairs $\mathcal{P} = \{(x_i^{pos}, x_i^{neg})\}$ where: (1) $x_i^{pos}$ = sample with higher desirability and (2) $x_i^{neg}$ = sample with lower desirability.

- The notion of "higher desirability" is task-dependent – (1) Regression: $y_{pos} > y_{neg}$ (e.g., higher engagement score) (2) Binary Classification: $x^{pos}$ from positive class, $x^{neg}$ from negative class

- For a dataset of size $N$, we can construct up to $\binom{N}{2}$ comparison pairs, though in practice we may sample a subset for computational efficiency.

**Advantages of Pairwise Formulation**: The pairwise framework offers two key advantages for automated feature discovery: (1) **Discriminative Focus**: By examining differences between $x^{pos}$ and $x^{neg}$, LLMs naturally focus on distinguishing characteristics rather than universally present attributes. This eliminates noise from features that appear equally in both successful and unsuccessful samples. (2) **Simplified Learning**: Each comparison $(x_i^{pos}, x_i^{neg})$ reduces complex preference learning to a focused binary discrimination problem: "What makes $x_i^{pos}$ better than $x_i^{neg}$?" This simplification enables clearer feature discovery compared to analyzing samples in isolation.

## 2.2 ALGORITHM INITIALIZATION

Before beginning the iterative process, FEST requires several initialization steps:

- **Data Preparation**: The input dataset $D$ is transformed into pairwise comparisons $\mathcal{P}$ and split into training and test sets $(D_{train}, D_{test})$.
- **Feature Bank**: An empty feature bank $F = \emptyset$ is initialized to store discovered features.
- **Feature Importance History**: A history tracker $H_{importance} = \emptyset$ is initialized to maintain feature importance scores across iterations for pruning decisions.
- **Batch Parameters**: Initial batch size $K_0$ and maximum batch size $K_{max}$ are set.
- **Convergence Criteria**: Accuracy threshold $\tau_{accuracy}$ for convergence and minimum importance threshold $\tau_{importance}$ for feature pruning are defined.
- **Optional Seed Features**: If domain knowledge is available, seed features can be provided to initialize the feature bank, otherwise the algorithm starts from scratch.

FEST processes comparison pairs in expanding batches to balance computational efficiency with feature quality. We begin with small batches (typically 50 pairs) and double the batch size each iteration until reaching a maximum threshold. This progressive expansion allows early iterations to discover fundamental patterns while later iterations capture more nuanced relationships. The expanding batch strategy serves two purposes: (1) it enables rapid initial discovery of high-impact features from limited data, and (2) it prevents computational bottlenecks that would arise from processing all pairs simultaneously in large datasets.

## 2.3 STAGE 1: PLAUSIBLE FEATURE DISCOVERY

For each comparison pair $(x_{positive}, x_{negative})$ in the current batch, we prompt large language models to generate plausible features explaining what differentiates the positive sample from the negative sample. We employ multiple prompt templates to encourage diverse feature discovery. This multi-perspective approach captures different aspects of the underlying patterns. Each pair generates $M$ plausible features per prompt template, creating a rich pool of potential features. The complete prompt templates are provided in Appendix G. The use of LLMs for plausible feature discovery leverages their natural language understanding to articulate human-interpretable explanations. Unlike traditional feature engineering, this approach automatically discovers relevant attributes without relying on scarce domain-specific expertise.

## 2.4 STAGE 2: FEATURE DEDUPLICATION

Stage 1 can produce thousands of plausible features, many expressing identical concepts through varied linguistic formulations. Left unchecked, this inflates the feature space and dilutes discriminative signal. We therefore perform semantic deduplication to group near-duplicates and consolidate them into representative summaries. We compute semantic embeddings of the plausible features conditioned on the task description using GritLM-7B Muennighoff et al. (2024). Unlike generic

text embeddings, these representations enable more accurate identification of conceptual similarity within the domain context. We then apply K-means to cluster semantically similar features and prompt an LLM to summarize each cluster into a single representative feature that preserves the core concept while filtering noise. This consolidation reduces dimensionality and computational cost in subsequent stages, and it amplifies true signal by unifying semantically equivalent features into a single representation.

## 2.5 STAGE 3: FEATURE RELEVANCE ASSESSMENT

This stage comprises of three steps: feature inference process where LLMs evaluate the presence or absence of each feature in $(x_i^{pos}$ and $x_i^{neg})$ samples, training a decision tree model on encoded features, and finally obtaining feature importance scores through the trained model for assessing their relevance.

**Feature Inference**: We split each comparison pair $(x_i^{pos}, x_i^{neg})$ into two individual samples and evaluate feature presence independently. For each individual sample $x$ and each feature $f_k$, the LLM evaluates whether feature $f_k$ is present or absent in sample $x$. This transforms our batch of $|B|$ comparison pairs into $2|B|$ individual samples ($|B|$ positive samples and $|B|$ negative samples). We define the feature inference function as:

$$g(f_k, x) = \begin{cases} 1 & \text{if feature } f_k \text{ is present in sample } x \\ 0 & \text{if feature } f_k \text{ is absent in sample } x \end{cases} \quad (1)$$

This feature inference process creates a feature matrix $\mathbf{X} \in \{0,1\}^{2|B| \times |F|}$ where rows represent the $2|B|$ individual samples (obtained by splitting $|B|$ comparison pairs) and columns represent features. Each element $X_{ij} = g(f_j, x_i)$ contains the binary presence indicator for feature $f_j$ in sample $x_i$. The corresponding label vector is $\mathbf{y} \in \{0,1\}^{2|B|}$ where $y_i = 1$ for positive samples ($x^{pos}$) and $y_i = 0$ for negative samples ($x^{neg}$).

**Decision Tree Classification**: Next, we train a decision tree classifier on the binary feature matrix to perform classification and evaluate feature relevance. The decision tree serves two critical purposes in our algorithm: (1) it learns discriminative patterns to for downstream classification tasks, and (2) it provides feature importance scores that guide feature bank evolution and pruning. We adopt decision trees for their interpretability, automatic feature selection, and ability to handle non-linear relationships between features. Once trained, the model is evaluated on the test set to measure classification accuracy, which determines algorithm convergence.

**Rule Bank Evolution and Pruning**: The feature bank evolves iteratively as new features are discovered and validated for predictive power. After each iteration, we extract feature importance scores from the trained decision tree and update the importance history $H_{importance}$ for each feature. This history tracker maintains importance scores across multiple iterations, enabling robust pruning decisions. Features with consistently low importance over the last three iterations (mean importance below $\tau_{importance}$) are removed from the feature bank. This pruning mechanism prevents unbounded growth while retaining only the most discriminative features, ensuring the model focuses on genuinely predictive patterns rather than noise.

The algorithm converges when either: (1) test accuracy exceeds a predefined threshold, indicating sufficient feature quality, or (2) all training data has been processed, ensuring comprehensive coverage. The detailed pseudocode is provided in Appendix **??**.

## 3 RESULTS AND DISCUSSIONS

### 3.1 DATASETS

We evaluate FEST across four categories of tasks to demonstrate its versatility and effectiveness in discovering interpretable features from observational data. We cover each of these four categories below:

1. **Social Science Tasks**:

- Headline Popularity (Zhou et al. (2024)): Predict which headline from a pair received the most clicks by users.
- Tweet Popularity (Tan et al. (2014)): Predict which tweet from a pair received more retweets.

- Deceptive Review Detection (Ott et al. (2013)): Predict if a given hotel review is genuine or deceptive.
- AI-Generated Content Detection (Liu et al. (2024)): Predict is a given story is human-written of AI-generated.
- Persuasive Argument Prediction (Pauli et al.): Predict which text from a pair is more persuasive.
- Mental Stress Detection(Turcan & McKeown (2019)): Predict if a given Reddit post indicates a state of stress.

2. **Language Understanding Tasks**: We adapt standard NLP benchmarks from GLUE (Wang et al. (2018)) to evaluate feature discovery on linguistic patterns:

- SST (Stanford Sentiment Treebank): Sentiment Analysis
- MRPC (Microsoft Research Paraphrase Corpus): Semantic Equivalence
- RTE (Recognizing Textual Entailment): Textual Entailment
- WNLI (Winograd NLI): Textual Entailment/reasoning

3. **Marketing Tasks**: We evaluate FEST on marketing data by extracting features that capture a brand's identity and using them to classify social media posts as on-brand or off-brand. A brand's identity is important for maintaining visibility, customer loyalty, and to stand out from competition Acar et al. (2024). We use a dataset of promotional social media posts of 12 brands (Khurana et al. (2025)) for this task. Results are averaged over all the brands. Evaluation is done per target brand, with posts labeled "on brand" if from the brand and "off brand" if from other brands in the same industry. The task (brand validation) is performed on both text and images.

4. **Synthetic Tasks**: To systematically evaluate the ability of models to perform compositional logical reasoning from text, we developed the **G**alactic **Lo**gical **R**easoning (**GLoRE**) benchmark. GLoRE is a synthetic, text-based environment designed to test a model's ability to deduce outcomes from complex logical rule relationships embedded in natural language. Each data point is a natural language description of a fictional alien species. The core of the benchmark lies in the eight ground truth logical rules that map the determinative features to galaxy preferences.

These rules include simple predicates (*e.g.*, $h_1$, $\neg h_2$) and complex compositional operators (*e.g.*, $h_1 \wedge h_2$, $h_1 \vee h_2$, $h_1 \oplus h_2$). The benchmark has eight balanced sub-datasets, one per logical rule, with positive samples satisfying the rule and negatives violating it (Table 2). Outcomes depend on logical feature combinations, not superficial cues (Appendix E). We evaluate binary classification: given an alien description and galaxy assignment, the model predicts **True** or **False**. Performance is measured with three key metrics. Detailed discussion for synthetic task is in E.5.

### 3.2 METRICS

We employ multiple evaluation metrics to assess different aspects of FEST's automated feature engineering performance. Different combinations of metrics are utilised for different experiments.

**1. Accuracy:** We first report accuracy as a primary measure of overall predictive performance across all tasks.

**2. Logical Consistency Score:** This metric evaluates if a model's predictions are internally consistent with the task's ground truth logic (e.g., AND, OR, XOR). The process first uses sentence embeddings (`Qwen/Qwen3-Embedding-4B`) to automatically identify the indices $(i^*, j^*)$ of the model-generated features that best align with our core hypotheses ($h_1, h_2$). For each sample $k$, we then use the boolean flags $(v_{k,i^*}, v_{k,j^*})$ associated with these features as inputs to the ground truth logical operator $\mathcal{R}$. The score is the fraction of samples where the operator's output matches the model's final prediction $p_k$. A high score indicates the model has learned the correct compositional structure of the problem as shown in 4.

$$\text{Logical Consistency} = \frac{1}{|\mathcal{D}|} \sum_{k \in \mathcal{D}} \mathbb{I}(p_k = \mathcal{R}(v_{k,i^*}, v_{k,j^*}))$$

**3. IoU Score:** This metric assesses the quality of the generated feature set ($\mathcal{G}$) against the ground truth set ($\mathcal{T}$), rewarding both correctness and conciseness. The intersection, $|\mathcal{T}_{\text{discovered}}|$, is the number of ground truth features with a semantically similar counterpart in $\mathcal{G}$, determined via an embedding

similarity threshold ($\tau = 0.7$). The IoU is the ratio of this intersection to the union of the two sets. A high score signifies a more accurate and efficient explanation from the model as shown in 3.

$$\text{IoU} = \frac{|\mathcal{T}_{\text{discovered}}|}{|\mathcal{T}| + |\mathcal{G}| - |\mathcal{T}_{\text{discovered}}|}$$

### 3.3 BASELINE METHODS

We compare FEST against the following baseline methods:

- **Zero-shot LLM**: Direct classification by an LLM with task description only, no examples provided.
- **Few-shot LLM**: LLM classification with task description and a few labeled examples in the prompt.
- **Features (Zero Shot) + LLM**: LLM discovers features from task description in zero-shot setting, then performs zero shot classification using these features.
- **Features (Few Shot) + LLM**: LLM discovers features from task description with few examples, then performs zero shot classification based on these features.
- **Features (Zero Shot) + Decision Tree**: LLM discovers features from task description in zero-shot, followed by decision tree classification. For classification, binary feature vectors are created for each sample. For a given sample's feature vector, the entry at the $i^{\text{th}}$ index indicates if the $i^{\text{th}}$ feature is present in that sample. This presence/absence is determined using LLMs.
- **Features (Few Shot) + Decision Tree**: LLM discovers features from task description with few examples, followed by decision tree classification.
- **Features (Zero Shot) + XGBoost**: LLM discovers features from task description in zero-shot, followed by XGBoost classification.
- **Features (Few shot) + XGBoost**: LLM discovers features from task description with few examples, followed by XGBoost classification.
- **Encoder**: BERT-based model (RoBerta ( Liu et al. (2019))) fine-tuned for classification on each text-based task. For the image task, embeddings were extracted from BLIP 2's Qformer (Li et al. (2023)) and a classification head was attached on top.

### 3.4 RESULTS

**1. Classification Performance:** Standard classification task for all the datasets. Performance assessed using accuracy. We report results using GPT-4o-mini (OpenAI (2023)) in table 1.

Table 1: Comparison of results obtained using GPT-4o-Mini. Best performing config is highlighted in bold, for each dataset. We treat the encoder as an oracle, and do not consider it while determining the best performing config, since it is very different from the broader theme of the rest of the approaches.

| Dataset Name | Model Name | LLM | | Features+LLM | | Features+DT | | Features+XGB | | Encoder | FEST |
|---|---|---|---|---|---|---|---|---|---|---|---|
| | | Zero Shot | Few shot | Zero Shot | Few shot | Zero Shot | Few shot | Zero Shot | Few shot | | |
| *Social Science Tasks* | | | | | | | | | | | |
| Deceptive Reviews | GPT | 0.514 | 0.656 | 0.604 | 0.612 | 0.676 | 0.634 | 0.680 | 0.642 | 0.920 | **0.696** |
| Headline Pairs | GPT | 0.593 | 0.587 | 0.550 | 0.589 | 0.535 | 0.552 | 0.520 | 0.533 | 0.490 | **0.612** |
| Retweet Pairs | GPT | 0.589 | 0.554 | 0.578 | **0.628** | 0.593 | 0.572 | 0.578 | 0.586 | 0.508 | 0.598 |
| GPT-generated content | GPT | 0.493 | 0.490 | 0.490 | 0.450 | 0.660 | 0.736 | 0.660 | 0.673 | 0.990 | **0.785** |
| LLama-generated content | GPT | 0.530 | 0.543 | 0.530 | 0.536 | 0.613 | 0.573 | 0.630 | 0.580 | 0.976 | **0.735** |
| Dreaddit | GPT | 0.626 | 0.704 | 0.662 | 0.650 | 0.688 | 0.720 | 0.716 | 0.720 | 0.722 | **0.745** |
| Persuasive Pairs | GPT | 0.718 | 0.662 | 0.760 | 0.722 | 0.738 | 0.738 | 0.724 | 0.744 | 0.518 | **0.875** |
| *Language Understanding Tasks* | | | | | | | | | | | |
| SST2 | GPT | 0.942 | 0.956 | 0.922 | 0.942 | 0.889 | 0.920 | 0.899 | 0.928 | 0.926 | **0.964** |
| RTE | GPT | 0.880 | 0.890 | **0.891** | 0.877 | 0.844 | 0.833 | 0.855 | 0.851 | 0.472 | 0.885 |
| MRPC | GPT | 0.698 | 0.718 | 0.703 | 0.699 | 0.788 | 0.759 | 0.792 | 0.759 | 0.869 | **0.844** |
| WNLI | GPT | 0.845 | 0.859 | 0.788 | 0.774 | 0.830 | **0.859** | 0.830 | 0.859 | 0.436 | 0.818 |
| *Marketing Tasks* | | | | | | | | | | | |
| Brand Validation (text) | GPT | 0.524 | 0.714 | 0.556 | 0.593 | 0.620 | 0.694 | 0.624 | 0.696 | 0.964 | **0.778** |
| Brand Validation (images) | GPT | 0.505 | 0.673 | 0.501 | 0.535 | 0.650 | 0.604 | 0.629 | | .673 | **0.713** |
| *Synthetic Tasks* | | | | | | | | | | | |
| H1 | GPT | 0.530 | 0.595 | 0.500 | 0.500 | 0.580 | 1.000 | 0.570 | 1.000 | 1.000 | **1.000** |
| H2 | GPT | 0.510 | 0.645 | 0.500 | 0.500 | 0.620 | **0.995** | 0.610 | 0.995 | 1.000 | 0.955 |
| H1 and H2 | GPT | 0.555 | 0.705 | 0.500 | 0.500 | 0.515 | 0.935 | 0.510 | **0.940** | 1.000 | 0.928 |
| H1 or H2 | GPT | 0.505 | 0.620 | 0.500 | 0.500 | 0.555 | 0.865 | 0.555 | 0.880 | 1.000 | **0.903** |
| H1 xor H2 | GPT | 0.470 | 0.495 | 0.500 | 0.500 | 0.500 | 0.540 | 0.505 | 0.540 | 1.000 | **0.583** |
| not H1 | GPT | 0.480 | 0.475 | 0.500 | 0.500 | 0.505 | 0.775 | 0.530 | 0.810 | 1.000 | **0.982** |
| not H2 | GPT | 0.490 | 0.575 | 0.500 | 0.500 | 0.640 | 0.755 | 0.640 | 0.755 | 1.000 | **0.925** |
| neither | GPT | 0.500 | 0.440 | 0.500 | 0.500 | 0.560 | 0.945 | 0.565 | **0.950** | 1.000 | 0.902 |

**2. Relationship Prediction and fidelity of discovered features:** We conduct this experiment using our synthetic environment. We use IoU to assess the intersection of generated features with ground truth features, and logical accuracy to measure correctness of the discovered relationship among features.

**3. Feature bank stability:** We measure feature bank stability via additions and deletions between iterations. EVOLVE (an adaptation of our Features (Few-shot) + XGBoost approach) iteratively updates the feature bank using 50 few-shot examples and the previous features, while FEST follows a similar procedure. As shown in 3, FEST's additions decrease and deletions increase over iterations, indicating pruning and a move towards convergence of features.

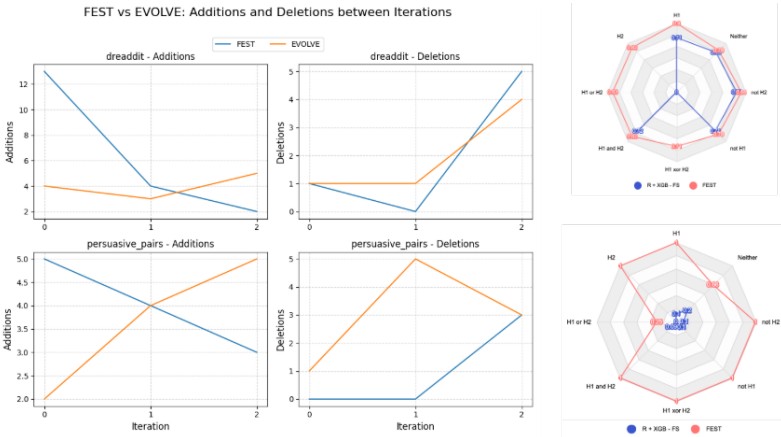

Figure 3: Comparison of additions and deletions for FEST and EVOLVE for Dreaddit and Persuasive Pairs.

## 3.5 DISCUSSION

From the results displayed in the previous section, we observe the advantages of FEST. In particular, FEST outperforms all other baselines (not accounting for the encoder) on most tasks, suggesting that it discovers high quality rules which enable accurate classification. Additionally, FEST outperforms all other methods in recovering ground truth features as well as their logical relationships on our synthetic benchmark. Finally, We observe that FEST provides more stable feature banks, with the features gradually converging. To show the interpretable nature of the features and decision rules discovered by FEST, we visualize the decision paths for two different samples of brand images, one on-brand and the other off-brand (1).

## 4 CONCLUSION

This paper addressed the long-standing trade-off between predictive performance and control in machine learning. We introduced **FEST** (*Feature Engineering with Self-evolving Trees*), a framework that automates feature engineering using large language models in combination with interpretable decision trees. FEST operationalizes a generate–deduplicate–validate loop that produces human-readable features, enabling practitioners to inspect and adjust decision rules while maintaining competitive accuracy. Empirical results across social science, NLP, marketing, and synthetic benchmarks show that FEST narrows the performance gap between interpretable and black-box models, while also providing feature sets that are stable, verifiable, and aligned with domain knowledge. These findings suggest that automated feature engineering can reduce reliance on manual expertise, scale more effectively across domains, and improve the practical usability of interpretable models. Future work should focus on improving scalability and integrating causal reasoning, as well as exploring FEST's potential in applications where actionable feedback or post-hoc explanations of black-box models are valuable. **Key takeaway:** FEST demonstrates that automated feature engineering can advance both accuracy and control, moving toward interpretable models that are practical at scale.

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

## A   RELATED WORK

### A.1   EARLY RULE-BASED AND MACHINE LEARNING SYSTEMS

The foundations of controllable machine learning can be traced to early rule-based systems and traditional ML approaches that prioritized interpretability. Samuel's checkers program Samuel (1959) exemplified this philosophy by using handcrafted board evaluation features like piece advantage and mobility that human experts could understand and validate. Similarly, expert systems like MYCIN Shortliffe (1976) relied on interpretable clinical features that physicians could trace through logical decision rules, achieving both reasonable performance and complete transparency in medical diagnosis tasks.

This feature-centric paradigm extended to practical applications throughout the 1980s and 1990s. Spam email classification systems used transparent features such as suspicious word frequencies ("free", "urgent", "click here") and sender metadata (domain reputation, header inconsistencies) to achieve both explainable decisions and reasonable accuracy Sahami et al. (1998). Text classification models relied on expert-designed features such as part-of-speech tags, n-gram statistics, and syntactic patterns Manning & Schütze (1999), providing clear pathways from input to prediction that domain experts could inspect and modify.

### A.2   TRADITIONAL MACHINE LEARNING MODELS

As machine learning matured, successive generations of algorithms maintained dependence on manual feature engineering while addressing specific algorithmic limitations. Support Vector Machines Cortes & Vapnik (1995) addressed the curse of dimensionality by mapping features into high-dimensional spaces where linear separation became possible, enabling effective classification even with limited training data. However, SVMs' performance remained critically dependent on appropriate feature selection and kernel choice, requiring domain expertise to achieve optimal results.

Decision trees Breiman et al. (1984) offered a fundamentally different approach, providing transparent hierarchical decision rules that could be directly interpreted by domain experts. Each internal node represented a simple threshold test on a single feature, making the entire decision process traceable and modifiable. This interpretability made decision trees particularly valuable in domains requiring regulatory compliance or expert validation, such as medical diagnosis and credit scoring Quinlan (1986). However, decision trees suffered from high variance and tendency to overfit, particularly with complex datasets containing many features or noise.

Ensemble methods emerged to address these limitations while preserving interpretability. Random Forests Breiman (2001) reduced overfitting by training multiple decision trees on bootstrap samples of the data and random subsets of features, then averaging their predictions. This approach maintained interpretability through feature importance measures while significantly improving predictive accuracy. Gradient boosting methods Friedman (2001) took a different approach, iteratively training weak learners to correct the errors of previous models. XGBoost Chen & Guestrin (2016) refined this concept with optimized implementations and regularization techniques, becoming dominant in structured data competitions while maintaining some degree of interpretability through feature importance scores.

Despite these algorithmic advances, all traditional ML methods shared a fundamental limitation: their success remained contingent on the quality of manually engineered features. As Domingos (2012) emphasizes, "Feature engineering is the key"—poor feature selection could render even sophisticated algorithms ineffective, while well-designed features could make simple models highly competitive.

### A.3  DEEP LEARNING AND ARCHITECTURAL REVOLUTION

The deep learning era (2012-present) fundamentally transformed machine learning by abandoning manual feature engineering in favor of automatically learning feature representations from raw data. Human expertise was redirected from feature design to architectural innovation, incorporating intelligent biases through model structure rather than explicit feature selection.

Convolutional neural networks LeCun et al. (2002) revolutionized image processing by learning hierarchical feature detectors directly from pixels, while recurrent neural networks and later transformer architectures Vaswani et al. (2017) achieved breakthrough performance in natural language processing by automatically extracting semantic representations from text. Attention mechanisms ? mimicked human selective attention by allowing models to focus on relevant input regions, while gated recurrent units ? incorporated memory mechanisms inspired by human cognition.

More recent innovations include debiasing techniques to remove societal stereotypes ?, reinforcement learning from human feedback (RLHF) to align model behavior with human preferences ?, and vision-language projection layers to bridge modalities Liu et al. (2023). This architectural revolution achieved dramatic performance gains across vision, language, and audio domains by trading control for performance—practitioners could no longer directly influence which features models emphasized or understand the reasoning behind predictions.

### A.4  POST-HOC EXPLAINABILITY METHODS

Recognizing the opacity of deep learning systems, the field developed post-hoc interpretability methods to explain model behavior after training. Techniques such as LIME Ribeiro et al. (2016), SHAP Lundberg & Lee (2017), and gradient-based attribution methods Selvaraju et al. (2017) emerged to provide insights into complex model decisions by attributing importance to input features or generating saliency maps.

However, these approaches face fundamental limitations that prevent them from restoring true control. They can produce misleading results when models exhibit poor generalization, when input features are highly correlated, or when complex feature interactions determine outcomes through non-linear relationships Adebayo et al. (2018). The problem of verifying explanations remains unresolved—there is no generally accepted methodology to establish whether these explanations faithfully capture the internal reasoning of a model Rudin (2019). Critically, post-hoc methods provide explanation without control: practitioners can obtain plausible explanations without gaining the ability to predict or systematically modify model behavior.

## A.5 LLM Reasoning Techniques

Large Language Models introduced reasoning techniques that appeared to bridge performance and interpretability. Chain-of-Thought (CoT) prompting Wei et al. (2022) demonstrated impressive performance improvements on reasoning benchmarks while appearing transparent through step-by-step explanations. Related techniques like self-consistency Wang et al. (2022) and tree-of-thoughts further enhanced reasoning capabilities.

However, systematic investigation reveals fundamental faithfulness problems with LLM reasoning. CoT can generate different reasoning paths for identical inputs across multiple runs, and explanations frequently diverge from the true computations driving predictions **?**. Recent analysis has fundamentally challenged the assumption that CoT provides genuine explainability, showing it is "neither necessary nor sufficient for trustworthy interpretability" Barez et al. (2025). Under complexity stress, reasoning models face "complete accuracy collapse beyond certain complexities" and "fail to use explicit algorithms" Shojaee et al. (2025), exposing CoT as sophisticated pattern matching rather than genuine reasoning.

## A.6 LLMs for Feature Engineering and Pattern Mining

Recent work has begun exploring how large language models can be used not just for prediction, but as engines for feature construction or pattern discovery. This is closely aligned with FEST's goal of using LLMs to generate interpretable, human-readable features. Below we summarize several representative approaches.

- HypoGenic: Zhou et al. (2024) introduced HypoGenic, which uses LLMs with multi-armed bandit approaches to generate and test hypotheses for social science applications.

- FeatLLM: LLMs for Tabular Feature Engineering. Han et al. (2024) introduce *FeatLLM*, which uses LLMs as feature engineers in few-shot learning settings with tabular data. FeatLLM generates candidate features through prompts and uses simple downstream models (e.g., linear regression) to select features. It significantly outperforms alternative LLM-based feature engineering methods, especially when training samples are limited.

- LLM-FE: Abhyankar et al. (2025) propose *LLM-FE* for tabular prediction tasks. In this framework, LLMs are combined with evolutionary search to generate feature transformation programs, guided by empirical performance feedback. This method incorporates domain knowledge and iteration, enabling discovery of novel transformations that improve performance over standard baselines.

- Embedding Domain-Specific Knowledge into Feature Pipelines: Batista (2025) explore using LLMs at the start of feature engineering pipelines to embed domain-specific knowledge. The idea is to seed the feature space with LLM-constructed feature combinations before applying evolutionary or symbolic methods. The approach tends to accelerate convergence and reduce computational load, though in many datasets it improves performance only modestly.

- FeRG-LLM: Ko et al. (2025) automates feature engineering via an LLM (fine-tuned) that generates features guided by reasoning and feedback loops. It demonstrates competitive or better performance than larger LLMs in classification settings while using fewer resources. FeRG-LLM is interesting because it emphasizes local interpretability (via the features it proposes) and deployability.

- Pattern Mining with ChatGPT: Weiss (2024) presents "An Exploration of Pattern Mining with ChatGPT", wherein the author describes an eight-step process combining human insight and LLM capabilities to discover patterns in data sources. Though not always focused on generating predictive features per se, this work illustrates how LLMs can help in extracting structured patterns and "rules" from data, which is relevant to FEST's goal of interpretable feature discovery.

**Comparison with FEST.** While these works offer valuable insights into LLM-driven feature construction, they often differ from FEST in important ways: many operate in fixed or narrow domains (e.g. tabular data only), rely heavily on downstream model performance without explicit "control" over feature semantics or thresholds, or limit themselves to generating features without iterative refinement or pruning. FEST aims to combine generation, deduplication, validation, and self-evolving decision trees to provide a more general, controllable, and scalable feature engineering solution.

## B  LIMITATIONS

While FEST demonstrates promising results for scalable, controllable modeling, it is important to acknowledge its limitations. We discuss them below:

**Dependence on LLM generations**: The quality of features proposed by FEST is bounded by the underlying LLM. Although LLMs capture broad domain knowledge, they may generate spurious, redundant, or semantically incoherent features. We mitigate this through deduplication and validation, but the framework inherits biases and blind spots present in the base model.

**Control is partial, not absolute**: FEST improves controllability relative to black-box models by producing human-readable features and decision rules. However, this control is limited to the space of features proposed and validated. Practitioners may still lack full guarantees of interpretability or complete transparency into why certain features were generated or selected.

**Correlation vs causation**: Features discovered by FEST should not be interpreted as causal without further analysis. Although our synthetic benchmark illustrates recovery of ground-truth rules, in real-world data the system may surface predictive correlations that do not correspond to causal mechanisms. Causal inference requires complementary methods beyond the scope of this work.

**Computational overhead**: FEST involves iterative feature generation, deduplication, and model retraining, which can be more computationally expensive than training a single predictive model. While these costs are amortized by reducing human labor in feature design, scaling to very large datasets or repeated runs may present challenges.

**Domain generalisation**: Our evaluation spans social science, NLP, and marketing datasets, along with synthetic benchmarks. While this demonstrates generality, we have not yet validated FEST in critical domains such as healthcare, finance, or scientific discovery, where requirements for reliability, causality and safety are higher.

Overall, these limitations point to directions for future work, including bias mitigation in LLM feature generation, integration with causal discovery methods and optimization for large-scale deployment.

## C  FUTURE WORK AND BROADER IMPACT

Beyond improving the scalability and causal grounding of FEST, we envision several exciting directions where the framework could extend its impact.

**Content optimization through actionable feedback**: FEST does not merely classify; it generates explicit, human-readable features that form the decision path for each prediction. This opens the door to applications such as optimizing headlines, tweets, or advertisements. For example, when distinguishing between engaging and non-engaging content, FEST can surface the exact linguistic or structural attributes that influence predicted engagement. Practitioners can then receive concrete feedback such as "headline length too short" or "absence of emotional keywords", allowing them to modify content in ways directly aligned with model logic. This shifts predictive modeling from passive forecasting toward active guidance.

**Post-hoc explainability of black-box models**: Another intriguing direction is to repurpose FEST as an explanation layer for opaque models. Suppose a neural network achieves state-of-the-art accuracy on a classification task. By labeling data with the network's predictions and then running FEST on top, one can extract interpretable features and decision rules that approximate the network's learned representations. This would combine the high performance of black-box models with FEST's ability to articulate insights in natural language, offering practitioners a window into otherwise inscrutable models.

**Scalability and causal discovery**: On the methodological side, future work should push FEST toward more efficient large-scale deployment and explore causal discovery. Enhancing the efficiency of the generate–deduplicate–validate loop will make FEST practical for massive datasets and near real-time applications. Integrating causal reasoning into the feature refinement process could help distinguish predictive correlations from genuine drivers of outcomes, a particularly critical need in scientific and policy domains.

Taken together, these directions suggest that FEST is not only a framework for automating feature engineering but also a step toward rethinking the role of models in human decision-making: from opaque predictors to transparent copilots that explain, advise and guide.

## D ALGORITHM PSEUDOCODE

Please take a look at the next page.

## E GALACTIC LOGICAL REASONING (GLoRE)

To rigorously evaluate the compositional reasoning capabilities of language models, we introduce the **Galactic Logical Reasoning (GLoRE)** benchmark. GLoRE is a synthetic, text-based environment designed to test a model's ability to deduce outcomes from complex logical rules embedded in natural language, while remaining robust to spurious correlations from irrelevant distractor attributes.

### E.1 DESIGN PRINCIPLES

The GLoRE benchmark is built upon the following core principles:

1. **Grounded determinative Factors:** The classification labels (galaxy preferences) are deterministically generated from a small set of underlying binary determinative features, rather than superficial textual patterns.

2. **Compositional Logic:** The benchmark is structured around a set of fundamental logical operations (AND, OR, XOR, NOT) applied to the determinative features, requiring models to perform multi-step, compositional reasoning.

3. **Controlled Distractors:** Alien descriptions contain several randomly sampled "distractor" features that have no determinative relationship with the outcome. This tests a model's ability to identify and isolate the true determinative variables.

4. **Natural Language Abstraction:** The underlying logical rules are not presented in a formal language but are abstracted away within varied, descriptive text, forcing models to reason over semantic content.

5. **Logically Constrained Negatives:** For each logical rule, negative samples are not chosen randomly. Instead, they are systematically generated from the precise set of feature combinations that falsify the rule, creating a challenging decision boundary.

**Generative Process**   Each data point in GLoRE consists of a textual description of a fictional alien species and a corresponding galaxy for which it has a preference.

**Alien Features.** We define two primary binary **determinative features**:

- **Telepathic Abilities ($h_1$):** A species is either telepathic ($h_1 = 1$) or non-telepathic ($h_1 = 0$).
- **Body Temperature Regulation ($h_2$):** A species is either warm-blooded ($h_2 = 1$) or cold-blooded ($h_2 = 0$).

To simulate a more realistic scenario where not all information is relevant, we introduce several **distractor features**, which are sampled uniformly at random:

- **Respiratory System:** e.g., "oxygen-based", "methane-based", "radiation-processing", "silicon-based".
- **Physical Structure:** e.g., "exoskeleton", "gaseous form", "crystalline structure".
- **Lifespan:** e.g., "short lifespan", "quasi-immortal", "cyclical life pattern".

**Textual Embedding.** The assigned determinative and distractor features are embedded into a narrative description using a set of diverse sentence templates. This ensures that the same underlying features can be presented with significant lexical and syntactic variation.

---

**Algorithm 1** FEST: Feature Engineering with Self-evolving Trees

---

1: **Input:** Dataset $D$ of $N$ samples with target scores
2: **Output:** Feature bank $F$ of validated features
3:
4: **Initialize:**
5: $F \leftarrow \emptyset$          ▷ Empty feature bank
6: $K \leftarrow K_0$          ▷ Initial batch size
7: $D_{paired} \leftarrow$ ConstructPairs($D$)          ▷ Create $\binom{N}{2}$ comparison pairs
8: $D_{train}, D_{test} \leftarrow$ Split($D_{paired}$)          ▷ Train-test split
9: $H_{importance} \leftarrow \emptyset$          ▷ Feature importance history
10: $idx_{current} \leftarrow 0$          ▷ Current batch index
11:
12: **while** not converged **do**
13:     $B \leftarrow$ SequentialBatch($D_{train}, K, idx_{current}$)          ▷ Sequential batch processing
14:
15:     **// Stage 1: Plausible Feature Discovery**
16:     $F_{raw} \leftarrow \emptyset$
17:     **for** each pair $(x_i^{pos}, x_i^{neg}) \in B$ **do**
18:        **for** each prompt template $t$ **do**
19:           $f \leftarrow$ LLM.PairwiseComparison($x_i^{pos}, x_i^{neg}$)          ▷ Generate $M$ plausible features
20:           $F_{raw} \leftarrow F_{raw} \cup f$
21:        **end for**
22:     **end for**
23:
24:     **// Stage 2: Deduplication**
25:     $C_{clusters} \leftarrow$ Cluster($F_{raw}, n_{clusters}$)          ▷ Group similar features
26:     $F_{batch} \leftarrow$ Summarize($C_{clusters}$)          ▷ Create cluster representatives
27:     $F_{current} \leftarrow F \cup F_{batch}$          ▷ Merge new features with existing feature bank
28:
29:     **// Stage 3: Feature Relevance Assessment**
30:     $\mathbf{X} \leftarrow$ LLM.EncodeFeaturePresence($B, F_{current}$)      ▷ $\mathbf{X} \in \{0,1\}^{2|B| \times |F_{current}|}$ ($|B|$ $x^{pos}$ + $|B|$ $x^{neg}$ samples)
31:     $\mathbf{y} \leftarrow$ SetTrueLabels($B$)          ▷ $\mathbf{y} \in \{0,1\}^{2|B|}$: 1 for $x^{pos}$, 0 for $x^{neg}$
32:
33:     **// Model Training and Evaluation**
34:     $\mathcal{M} \leftarrow$ DecisionTree.Train($\mathbf{X}, \mathbf{y}$)
35:     $acc \leftarrow$ DecisionTree.Evaluate($D_{test}, F_{current}$)
36:
37:     **if** $acc \geq \tau_{accuracy}$ **then**
38:        **break**          ▷ Convergence achieved
39:     **end if**
40:
41:     **// Feature Pruning and Batch Update**
42:     $\mathbf{I} \leftarrow$ DecisionTree.GetFeatureImportance()
43:     $H_{importance} \leftarrow$ UpdateHistory($H_{importance}, \mathbf{I}$)          ▷ Maintain feature importance history
44:     $F \leftarrow$ PruneFeatures($F_{current}, H_{importance}, \tau_{importance}$) ▷ Remove low importance features
45:     $K \leftarrow \min(2K, K_{max})$          ▷ Double batch size
46:     $idx_{current} \leftarrow idx_{current} + K$          ▷ Update batch index
47: **end while**
48:
49: **return** $F, \mathcal{M}$

---

**Task Formulation and Dataset Structure**    The core task in GLoRE is a set of binary classification problems. For each logical rule, the model is given an alien's description and must predict whether the species prefers the galaxy associated with that rule. The benchmark is organized into 8 distinct sub-datasets, each corresponding to a specific ground truth rule (Table 2). Each sub-dataset contains 1,000 examples, perfectly balanced with 500 positive and 500 negative instances.

- **Positive Samples:** An alien description whose features satisfy the logical rule $\mathcal{R}$.
- **Negative Samples:** An alien description whose features explicitly satisfy $\neg\mathcal{R}$. For instance, for the rule $h_1 \wedge h_2$, negative samples are constructed from the feature sets $\{h_1 = 1, h_2 = 0\}$, $\{h_1 = 0, h_2 = 1\}$, and $\{h_1 = 0, h_2 = 0\}$.

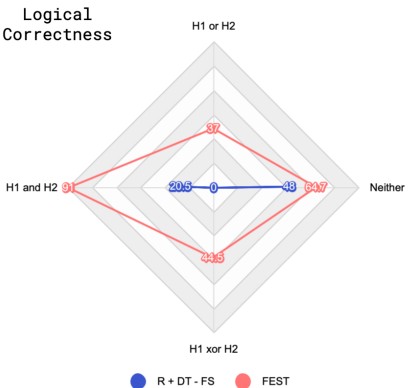

Figure 4: The plot shows logical consistency score between the strongest baseline Rules + Decision Tree (R + DT) and FEST. The zeros in the plot depicts the baseline rules were not able to exceed similarity score threshold.

**Example Data Point (Category C4: XOR)**    Consider the task for rule $h_1 \oplus h_2$.

**Positive Example (`label:   1`):**

- *Determinative Features:* $h_1 = 1$ (telepathic), $h_2 = 0$ (cold-blooded).
- *Description:* "The Xan'konar is a crystalline structure... It **exchanges information through mental connection**... this organism **relies on external heat sources to regulate temperature**."
- *Reasoning:* The species satisfies the XOR condition.

**Negative Example (`label:   0`):**

- *Determinative Features:* $h_1 = 1$ (telepathic), $h_2 = 1$ (warm-blooded).
- *Description:* "Known for its distinctive amorphous structure, the Mev'orian **communicates directly through thoughts**... This species... **maintains constant internal body temperature**."
- *Reasoning:* The species violates the XOR condition as both features are true.

This controlled setup allows for a precise evaluation of a model's ability to learn and apply compositional logical rules from textual data.

### E.2    SYNTHETIC EXPERIMENTS AND EVALUATION

To assess the logical reasoning capabilities of the models, we conduct a series of experiments on our synthetic **GLoRE** benchmark. We first reformulate the task into a binary classification problem and then evaluate the models using a suite of metrics designed to measure both predictive accuracy and the internal logical consistency of their reasoning.

Table 2: The 8 ground truth rules for Alien Species' Galaxy Preferences governing the GLoRE sub-datasets. Each rule maps a specific set of determinative features to a preferred galactic environment.

| Logical Rule $\mathcal{R}$ | Description | Galaxy Preference |
|---|---|---|
| **H1** | Aliens with telepathic abilities. | Low stellar density. |
| **H2** | Aliens that are warm-blooded. | High radiation. |
| **H1 $\wedge$ H2** | Aliens that are both telepathic and warm-blooded. | Low stellar density + High radiation. |
| **H1 $\vee$ H2** | Aliens that are telepathic or warm-blooded. | Variable/adaptive environments. |
| **H1 $\oplus$ H2** | Aliens exclusively telepathic or warm-blooded, not both. | Specialized/targeted environments. |
| **¬H1** | Aliens that are not telepathic. | High stellar density. |
| **¬H2** | Aliens that are cold-blooded. | Low radiation. |
| **¬H1 $\wedge$ ¬H2** | Aliens that are neither telepathic nor warm-blooded. | High density + Low radiation. |

### E.3 TASK FORMULATION

We re-model the dataset as a validation task. For each data point, the alien description and a potential galaxy assignment are combined into a single input sentence. The model's task is to perform binary classification, predicting whether the assignment is **True** (i.e., the alien's features are logically consistent with the galaxy's requirements) or **False** (i.e., the assignment is logically inconsistent).

**Justification.** This binary classification setup is crucial for a rigorous evaluation. It transforms the problem from a potentially ambiguous generative task into a well-defined discriminative one. This allows for direct and quantifiable measurement of the model's ability to apply the underlying logical rules, providing a clearer signal of its reasoning fidelity compared to open-ended generation.

### E.4 EVALUATION METRICS

We evaluate model performance on our synthetic tasks using three distinct metrics.

**1. Standard Accuracy.** This is the primary measure of predictive performance. It is calculated as the proportion of all samples that are correctly classified as either True or False.

$$\text{Accuracy} = \frac{\text{Number of Correct Predictions}}{\text{Total Number of Predictions}}$$

**2. Logical Consistency Score.** This metric goes beyond surface-level accuracy to evaluate whether the model's predictions are consistent with the ground truth logical operations (*e.g.*, AND, OR, XOR). The evaluation is a two-step process designed to verify the model's internal reasoning chain.

**Step 1: Automated Rule Identification.** For a given task (e.g., C2: $h_1 \wedge h_2$), the model under evaluation generates a set of natural language rules, $\mathcal{G} = \{g_1, ..., g_N\}$. We must first identify which of these generated rules correspond to our ground truth hypotheses, $h_1$ (telepathic) and $h_2$ (warm-blooded). To achieve this without manual intervention, we use a powerful sentence-embedding model, `Qwen/Qwen3-Embedding-4B`, to encode both the generated rules and our ground truth rule descriptions. We then compute the cosine similarity between the embeddings. The two generated rules with the highest similarity to our ground truth descriptions for $h_1$ and $h_2$ are selected, provided their similarity scores exceed a threshold of $\tau = 0.7$. Let their indices be $i^*$ and $j^*$.

**Step 2: Truth Table Verification.** For each sample in the dataset, the model provides a boolean vector $\mathbf{v} \in \{0, 1\}^N$, where each element $v_k$ indicates whether the sample satisfies the generated rule $g_k$. Using the indices $i^*$ and $j^*$ identified in Step 1, we extract the boolean values $(v_{i^*}, v_{j^*})$. These values serve as inputs to the ground truth logical truth table for the task. For example, for the XOR task (C4), the expected output is $v_{i^*} \oplus v_{j^*}$. The **Logical Consistency Score** is the accuracy of the model's final predictions when compared against this expected output from the truth table.

$$\text{Logical Consistency} = \frac{1}{|\mathcal{D}|} \sum_{k \in \mathcal{D}} \mathbb{I}(p_k = \mathcal{R}(v_{k,i^*}, v_{k,j^*}))$$

where $p_k$ is the model's prediction for sample $k$, $\mathcal{R}$ is the ground truth logical operator (e.g., $\wedge, \vee, \oplus$), and $\mathbb{I}$ is the indicator function. This metric provides a strong signal of whether the model has learned the correct compositional structure of the task.

**3. IoU Score.** While the Logical Consistency Score validates the model's predictions against its self-generated rules, it does not directly evaluate the quality of the rule set itself. To address this, we introduce the Intersection over Union (IoU) score. This metric assesses a model's ability to generate a set of rules that is both **correct** (covering all ground truth principles) and **concise** (excluding irrelevant or redundant statements).

Let $\mathcal{T} = \{t_1, ..., t_{|\mathcal{T}|}\}$ be the set of ground truth rules for a task, and $\mathcal{G} = \{g_1, ..., g_{|\mathcal{G}|}\}$ be the set of rules generated by the model. We define the set of discovered ground truth rules, $\mathcal{T}_{\text{discovered}}$, as the subset of $\mathcal{T}$ where each rule $t_i$ has at least one matching rule $g_j \in \mathcal{G}$. A match is determined if the cosine similarity of their sentence embeddings, $\text{sim}(E(t_i), E(g_j))$, exceeds a threshold of $\tau = 0.7$.

The IoU score is then calculated as the size of the intersection (the number of discovered rules) divided by the size of the union of the two sets:

$$\text{IoU} = \frac{|\mathcal{T}_{\text{discovered}}|}{|\mathcal{T}| + |\mathcal{G}| - |\mathcal{T}_{\text{discovered}}|}$$

A high IoU score indicates that the model has successfully recovered the complete set of ground truth principles without generating many extraneous rules, signifying a more accurate and efficient explanation of its reasoning.

### E.5 RESULTS DISCUSSION

We evaluate our approach on a series of synthetic tasks designed to test the model's ability to handle fundamental logical compositions. The results demonstrate a clear advantage for our proposed method, FEST, over baseline approaches and highlight specific weaknesses in standard large language models.

First, we assessed the accuracy of a GPT-based model on tasks involving combinations of two underlying hypotheses, $H_1$ and $H_2$. The model achieves near-perfect accuracy on identifying the simple hypothesis $H_1$ (1.000) and its negation $\neg H_1$ (0.982). Performance remains strong for other logical operations, including conjunction ($H_1 \wedge H_2$, 0.940), disjunction ($H_1 \vee H_2$, 0.903), and the neither case ($\neg(H_1 \vee H_2)$, 0.950). However, the model's performance deteriorates dramatically for the exclusive disjunction ($H_1 \oplus H_2$), where accuracy falls to 0.583, indicating a significant difficulty in reasoning about this type of logical constraint, performing only slightly better than random chance.

When evaluating the logical consistency of generated explanations, FEST shows a substantial improvement over the `R + DT - FS` baseline. For the conjunction task ($H_1 \wedge H_2$), FEST achieves a consistency score of 91, compared to the baseline's 20.5. The baseline's limitations are particularly pronounced for disjunctive statements; it scores 0 for both inclusive ($H_1 \vee H_2$) and exclusive disjunction ($H_1 \oplus H_2$), signifying a complete failure to produce logically coherent outputs for these cases. In contrast, FEST maintains respectable scores of 37 and 44.5, respectively.

We further measure the quality of feature attribution using Intersection over Union (IoU) and Cosine Similarity against a stronger `R + XGB - FS` baseline. In the IoU evaluation, FEST consistently identifies the correct feature set with perfect or near-perfect accuracy, achieving a score of 1.0 for $H_1$, $H_2$, their conjunction, exclusive disjunction, and their individual negations. The baseline, however, struggles immensely, with scores often at or near zero, such as 0.0 for $H_2$ and $H_1 \oplus H_2$. While FEST's IoU score for inclusive disjunction ($H_1 \vee H_2$) is lower at 0.25, it still provides a meaningful signal compared to the baseline's score of 0.

The results for cosine similarity reinforce this conclusion. FEST consistently yields higher similarity scores, indicating that its generated explanations are more semantically aligned with the ground truth concepts. For example, on the simple hypothesis $H_1$, FEST achieves a similarity of 0.9, substantially higher than the baseline's 0.71. This advantage persists across all logical compositions. For instance, in the case of $\neg H_2$, FEST scores 0.842, demonstrating a more precise semantic understanding than the baseline's 0.786.

Collectively, these experiments on synthetic data underscore FEST's superior ability to generate explanations that are not only accurate but also logically consistent and semantically faithful to complex logical concepts.

## F LLM USAGE

Large Language Models were used solely as a writing assistance tool during the preparation of this manuscript. Specifically, LLMs were employed to: (1) Polish and refine the language and clarity of written sections (2) Assist with formatting and organization of content. LLMs were *not* involved in any aspect of the research ideation, methodology design, experimental design, data analysis or interpretation of results.

## G PROMPT TEMPLATES

Please refer to the code provided in the supplementary materials.

## H HYPERPARAMETERS

For reproducibility, we summarize the hyperparameters used in FEST below:

- **Batch Processing**: Initial batch size $K_0 = 50$ pairs, maximum batch size $K_{max} = 200$ pairs, batch size doubles each iteration until reaching maximum.
- **Feature Generation**: $M = 5$ features generated per comparison pair per prompt template, using 2 prompt templates (positive and negative framing).
- **Semantic Clustering**: K-means with $k = 30$ clusters for feature deduplication, using GritLM-7B embeddings with task-conditioned prompts.
- **Feature Filtering**: Minimum importance threshold $\tau_{importance} = 0.05$ for feature pruning, importance history tracked over last 3 iterations.
- **Convergence**: Accuracy threshold $\tau_{accuracy} = 0.8$ for early stopping, maximum 10 iterations if convergence not reached.
- **Decision Trees**: Scikit-learn DecisionTreeClassifier with default parameters, random state fixed for reproducibility.
- **LLM Configuration**: Temperature = 0.1 for feature generation, temperature = 0.1 for feature inference to ensure deterministic binary outputs.

## I SAMPLE FEATURES

We present some features discovered by FEST for stress detection (Dreaddit);

- "Select the post that expresses overwhelming emotions because it indicates emotional instability and significant stress due to lack of regulation.",
- "Select the post that expresses agency because it conveys personal control, choice, and a sense of efficacy over circumstances.",
- "Select the post that references physiological issues like sleep disturbances, exhaustion, and concerns about heart health."
- "Select the post that expresses urgency in seeking immediate help and support with phrases like 'I need help'."
- "Select the post that lacks clear articulation because it indicates confusion and cognitive disorganization, suggesting underlying distress."
- 

