# OpenReview forum: "Scaling Controllable Modeling Via Self-Evolving Feature Engineering"
_ICLR.cc/2026/Conference — ICLR 2026 Conference Withdrawn Submission_

### Official Review · Reviewer_bQrm · 2025-10-28

**Soundness:** 2
**Presentation:** 2
**Contribution:** 1
**Rating:** 4
**Confidence:** 3

**Summary:**

This paper proposes FEST, a framework that automates feature engineering by leveraging LLMs to produce features for interpretable decision trees. FEST aims to address the trade-off between model performance and control. The framework proceeds in three iterative stages: 1. Plausible feature discovery, which uses LLMs to generate candidate features via pairwise comparisons of positive and negative samples. 2. Feature deduplication, which clusters semantically similar features using K-means and summarizes the cluster into a new feature. 3. Feature relevance assessment, which evaluates features with decision trees to construct a refined feature bank. The framework is tested on different tasks, including social science, NLP, marketing, and a proposed synthetic logical reasoning benchmark. FEST achieves competitive or superior performance to zero/few-shot LLMs baselines, and some other baselines.

**Strengths:**

- The combination of decision trees with LLMs can mitigate the transparency and controllability lacking in the deep learning models.

- The framework reduces human involvement for manual labeling.

- Experimental datasets are diverse, across multiple tasks (social science, NLP, marketing) and the proposed synthetic reasoning task.

**Weaknesses:**

Reliability of the feature inference step. Claim of control. Comparison to more sophisticated LLM baselines. See questions below

**Questions:**

- How reliable is the feature inference step? The third stage of FEST assumes that LLMs can reliably infer the presence or absence of intermediate features for each sample. However, this step seems inherently challenging: if the LLM itself cannot accurately predict the final task label, how can we be confident that it can correctly infer the intermediate features that are causally or semantically related to that label?
	- How to ensure that LLM feature inference is accurate and consistent across samples?
	- Are there any quantitative checks to validate the correctness of these inferred features? Without such checks, how do we distinguish between genuinely informative intermediate features and artifacts of LLM bias?

- The paper claims that FEST allows better control. Does this mean practitioners can modify features or decision rules to observe how such an intervention alters the prediction results? Could the authors demonstrate this empirically, for example, by altering a subset of feature labels or editing rules to observe consistent and predictable changes in predictions? Such an experiment would concretely validate FEST's controllability claim.

- Any comparison to the chain of thought baselines? Using proper CoT reasoning can potentially improve LLM performance a lot.

---

### Official Review · Reviewer_vSZR · 2025-10-29

**Soundness:** 1
**Presentation:** 1
**Contribution:** 2
**Rating:** 2
**Confidence:** 4

**Summary:**

This paper presents an algorithm called FEST: Feature Engineering with Self-evolving Trees, which is designed to automatically extract tabular features from textual datasets and incorporate them into a simple and interpretable decision tree to be used for downstream classification tasks. This approach is implemented by creating a dataset of positive/negative pairs, which an LLM is then asked to distinguish. The reasons given for pairs being different are clustered and formed into a smaller set of non-redundant rules. Once these rules are obtained, the authors use an LLM to apply the rules to new data and form a tabular dataset, which a decision tree can be fit on. The feature importances from these trees are used to guide further search by pruning features which are not important to the decision problem.

**Strengths:**

1. The system seems to be well-engineered, in the sense that the pipeline for feature extraction and evaluation is sensible, addresses redundancy and feature relevance, and the result is indeed a decision tree with LLM-extracted features that are hopefully relevant and interpretable.

2. The synthetic task GLOrE could be useful to future researchers trying to investigate feature extraction approaches on a baseline with known features.

**Weaknesses:**

1. It is unclear what the unique contribution of this work to the literature is. The related work section of the appendix needs to be moved to the main body of the paper and expanded upon significantly.

2. The introduction and related work reads as an unnecessary history of the entire arc of machine learning, and does not situate the paper in relevant literature.

3. The paper is unfortunately quite poorly organized and, frankly, poorly written. There are consistent grammatical errors, spelling errors, and punctuation mistakes. The language is flowery, imprecise, and unscientific in nature.

4. Many assumptions and important research decisions are either made implicit or left unverified and unjustified. For one example of many, the authors' justification for using pairwise comparisons (a core component of their method) is a reference to "Psychology research" at large, with one citation from a 1927 paper justifying the entire idea. There is no discussion of alternative approaches, nor a discussion of the relevant research, nor a justification for why 100-year old psychology research should be applicable to an LLM.

5. The evaluation of this method is severely lacking. At a minimum, the evaluation needs 1) repeated runs of each trial, with means and standard deviations reported; 2) analysis using a variety of LLM backbones instead of just GPT-4o-mini; 3) comparison against existing state-of-the-art methods, instead of contrived baselines.

6. The experimental results are also very limited -- The feature bank stability results are shown for only two datasets and only two iterations each, without any analysis of the effect of different hyperparameters or other choices. They also compare against a contrived baseline EVOLVE.

7. There are missing citations all over the Appendix (?'s instead of citations).

**Questions:**

# Questions

I do not have questions at this time; I think that the weaknesses I have described summarize my opinion, which answers to questions will not change.

# Suggestions

1. I would recommend that the authors remove the sections detailing the history of machine learning from the introduction and the appendix, as they do not contribute to the intellectual novelty of the idea.

2. I refer the authors to the book, "The Elements of Style," by White and Strunk. Reading this book will help the authors to write prose that is direct, concise, and easier to read and understand. Incorporating the advice from this book would improve the quality of this paper dramatically. Related to this; the use of LLMs as a writing aid can be helpful in moderation, but the language in this paper has the distinct feel of over-explanation, wordiness, and exaggeration commonly ascribed to LLMs.

3. Be much more thorough in the experimental evaluation and in the comparison against other methods in the literature. Avoid using contrived baselines, and be sure to report statistics over repeated trials of all experiments where it's possible.

---

### Official Review · Reviewer_nAM2 · 2025-11-01

**Soundness:** 3
**Presentation:** 3
**Contribution:** 3
**Rating:** 6
**Confidence:** 3

**Summary:**

This paper introduced FEST (Feature Engineering with Self-evolving Trees), which is a novel framework to tackle both control and performance. It also introduced GLoPE, a controlled synthetic benchmark to test a model's ability to deduce outcomes.  FEST integrates large language models (LLMs) as feature discovery engines and combines them with interpretable decision trees. It uses LLMS for feature generation, does semantic clustering and deduplication using embedding similarity and finally feature validation and pruning through decision tree relevance scores. The results show that this novel approach in FEST can achieve competitive numbers in comparison with blackbox models.

**Strengths:**

1. The paper discusses an important problem in the domain of machine learning.
2. The introduced framework in FEST seems sound, where LLMs ae used for discriminative feature discovery via pairwise comparisons and classical ml-based approaches for the rest.
3. The methodology (along with problem formulation) is clear. Multiple baselines were used to compare the results with a strong experimental setup.
4. The results show that FEST can recover determinative features (considering GLoRE)

**Weaknesses:**

1. Feature 3 seems to be not clear. Please fix that.
2. It would be better to provide some quantitative numbers of results in the abstract and the introduction (may be averages). I had to read till the results section (3.4) to find the numbers.
3. The introduction section seems a bit too long. Maybe the authors can make it shorter and have a simplified version of the FEST algorithm (Algorithm 1) in the main paper.
4. It is possible that LLM based feature inference to have a significant overhead in terms of resource utilization. More discussion on that and deployment in a real-world setup is encouraged.
5. LLMs can both hallucinate and generate redundant features at times. As explained in the paper, clustering can be a potential solution. However, providing more evidence on this is encouraged.

**Questions:**

1. How scalable is FEST when applied to large-scale high-dimensional datasets? How does the computational cost of LLM-based feature inference compare to that of classical interpretable models?
2. Can you elaborate on how human practitioners can intervene in or interpret FEST’s evolving feature bank?
3.  GLoRE benchmark demonstrates FEST’s ability to recover determinative features. Can you explain how well this translates to more complex (real-world scenarios) data where ground-truth logical structures are unknown?
4. Would it be possible to explore other open-source LLMs and see performance? Maybe LLMs with limited parameters?

---

### Official Review · Reviewer_4YXx · 2025-11-05

**Soundness:** 2
**Presentation:** 3
**Contribution:** 3
**Rating:** 2
**Confidence:** 4

**Summary:**

The paper introduces a method, FEST, for automated feature engineering using a combination of models and algorithms. The method iteratively develops features in three stages. The first uses an LLM to output potential features (as human-readable text) based on pairwise comparisons between samples. The second generates embeddings for the potential features using an LLM, clusters the embeddings, and summarizes each cluster with an LLM. These summaries become the features. The final stage uses an LLM to make a binary decision about if each feature is present in each sample, trains a decision tree on these features, and retains the features which score high in importance. The iterative process ends when the decision tree achieves a certain level of accuracy on a prediction task. FEST enables the extraction of features, encoded as human-readable text, and trains a decision tree using those features. By automatically developing features which are human-readable text and training decision trees, FEST offers a highly interpretable method while requiring no human domain expertise.

**Strengths:**

Originality: I’m not an expert in feature engineering but it does seem quite novel to use language models and decision trees, together, to iteratively produce features.

Quality: While I have significant concerns about the benchmarks they use, it’s excellent that they included a benchmark which uses images, not just text. They also use numerous benchmarks and show consistent performance.

Clarity: The work clearly lays out a fairly complex method. More detail is needed for multiple parts of the paper, but some is provided in the Appendix.

Significance: Interpretable methods deserve more attention and many domains would benefit from a method which offers these kind of easily understandable features and predictions.

**Weaknesses:**

## Baseline selection
If the goal is to scale beyond what humans can do, that should be a baseline. And does it surpass human domain expertise? Choosing a baseline that does have hand crafted features to see how what your method finds compares would be stronger evidence.

They could have experimented with humans replacing the LLMs at different steps, as a kind of ablation study. Can humans write better natural language features than an LLM (ie replace step 1+2 with a human) for a given dataset, especially if they are domain experts? This would make much more sense than most of their baselines, all of which use LLMs for feature detection and/or classification.

How would their method compare to the tabular-data-focused approaches they cite in their Related Work section? Their method should work for that (LLMs can process tabular data) and if it doesn’t, that would be useful to know

I would like to see an evaluation of the quality of outputs at each step in the algorithm, since errors can compound across the steps especially when embedding and clustering. Does something get “lost” at that stage; how to validate that it only deduplicates?

It would be stronger to compare their method to state of the art on each of these benchmarks, rather than treating finetuning RoBerta as such.

## Limitations section
There is no limitations section in the main body. Perhaps the introduction could be shortened a bit in order to touch on limitations.
The limitations section they do have, in the Appendix, is very brief, even though I would expect there to be quite interesting empirical limitations to discuss. When their method doesn't beat a baseline (as it does vs some of the baselines), why? Do the instances of lower performance reveal any limitations of the method?

## Benchmarks
The GLoRE contribution is unclear. I can guess at what the benchmark is trying to demonstrate in terms of the method’s performance, but the paper doesn’t have room for this to be properly introduced. Perhaps it should be dropped. It may be better to focus in on a few benchmarks and explore the results in more depth, for example by looking at samples where their method and the baseline method had the same or different errors, to theorize why.

While it is good to show the method generalizes to a degree, I believe the paper errs too much on the side of breath when more depth - an analysis of the performance on a particular benchmark - would make their claims stronger.

They compare feature stability to a baseline called “EVOLVE” but it isn’t fully explained nor justified as the choice of baseline. If this is an existing method, minimally a citation is needed.

## Conceptual
* Control is not synonymous with interpretability or explainability. Being able to understand features or what factored into a prediction does not necessarily mean the method can be intervened on to change this, especially when their method is actually somewhat complex and involves multiple LLMs, which are not very controllable.

* Are features generated by language models actually more interpretable?

A sample feature from the Appendix is “Select the post that lacks clear articulation because it indicates confusion and cognitive disorganization, suggesting underlying distress.” - this is a high level of abstraction, how do you know if a post “lacks clear articulation”? In their method, an LLM makes this classification. The feature appears interpretable, but whether or not a sample has this feature is an entirely black-box prediction from an LLM.

* Are features best expressed in natural language?

For example, when classifying images, features might be better expressed in interpretable but numeric values like the distribution of color values or measures of lightness/darkness.

## Minor edits
045: drop “such as healthcare diagnostics, financial lending, and policy decisions” since it’s in the next sentence

153: Verifiable -> interpretable (or traceable? transparent?) - “verifiable” has very specific connotations related to proofs

Figure 2: It look me awhile to figure out what the righthand side was. Perhaps the caption could state that the lefthand side outlines the algorithm and the righthand side shows it using an example.

297: “it learns discriminative patterns to for downstream” typo

312: broken link to Appendix

326: “Predict is a given story is human-written of AI-generated.” typos

Table 1:
* Put that it’s a measure of accuracy in table caption
* “GPT” is not a model name - put the specific model used

411: “We report results using GPT-4o-mini” - what aspect did they use GPT-4o for? Like it formatted the table?

462: “We” shouldn’t be capitalized

Figure 3 is basically too small to read (use SVG images?)

Appendix has broken references

**Questions:**

In summary, I feel that perhaps the paper tries to squeeze in too much. Focusing in on a deeper analysis of key benchmarks and baseline methods would give stronger evidence and a clearer picture of the strengths and weaknesses of the method. The introduction and motivation for pairwise comparison could both be shortened to give authors more space.

Some points I would like to see addressed:
* Could the limitations be moved to the main body of the paper and address some of the issues raised in the weaknesses section above?
* Could some kind of comparison to human engineered features be included? Could they compare the performance of their method on tabular data to the results of the methods cited in related work?

I recognise the introducing new benchmarks at this stage is difficult. If that is not possible, could a deeper analysis of the classification performance be included?

A minor point but something that remains unclear - the paper claims to be focused on feature engineering, but as far as I understand, it actually introduces both a way of engineering features as well as using them in a decision tree. Sometimes it seems like their claims are just about the challenges of feature engineering, other times about interpretable methods (both features and models) as whole. If this is just about _features_, I would be interested to see how their feature bank performs when used by a different model (not a decision tree). This would show whether their method produces useful features in a general sense, or just in conjunction with the co-developed decision tree. Or, perhaps you could argue that in practice, features are always developed for a particular kind of model, so this is expected. Curious what the authors would say on this topic.

---

### Note · Authors · 2025-12-02

I have read and agree with the venue's withdrawal policy on behalf of myself and my co-authors.